

# Exploring the Parameters Space of the Regional Climate Model COSMO-CLM 5.0 for the CORDEX Central Asia Domain

Emmanuele Russo[1,2,3], Silje Lund Sørland[4], Ingo Kirchner[1], Martijn Schaap[5,1], Christoph C. Raible[2,3], and Ulrich Cubasch[1]

[1]Institute for Meteorology, Freie Universität Berlin, Carl-Heinrich-Becker-Weg 6-10, 12165, Berlin, Germany
[2]Climate and Environmental Physics, Physics Institute, University of Bern, Sidlerstrasse 5, 3012, Bern, Switzerland
[3]Oeschger Centre for Climate Change Research, University of Bern, Hochschulstrasse 4, 3012, Bern, Switzerland
[4]Institute for Atmospheric and Climate Science, ETH Zurich, Universitätstrasse 16, 8092 Zurich, Switzerland
[5]TNO Built Environment and Geosciences, Department of Air Quality and Climate, Princetonlaan 6, 3584, CB, Utrecht, The Netherlands

**Correspondence:** russo@climate.unibe.ch

**Abstract.** The parameter uncertainty of a climate model represents the spectrum of the results obtained by perturbing its empirical and unconfined parameters used to represent sub-grid scale processes. In order to assess a model reliability and to better understand its limitations and sensitivity to different physical processes, the spread of model parameters needs to be carefully investigated. This is particularly true for Regional Climate Models (RCMs), whose performances are domain-
5   dependent.

In this study, the parameter space of the RCM COSMO-CLM is investigated for the CORDEX Central Asia domain, using a Perturbed Physics Ensemble (PPE) obtained by performing 1-year long simulations with different parameter values. The main goal is to characterize the parameter uncertainty of the model, and to determine the most sensitive parameters for the region. Moreover, the presented experiments are used to study the effect of several parameters on the simulation of selected
10   variables for sub-regions characterized by different climate conditions, assessing by which degree it is possible to improve model performances by properly selecting parameter inputs in each case. Finally, the paper explores the model parameter sensitivity over different domains, tackling the question of transferability of an RCM model setup to different regions of study.

Results show that only a sub-set of model parameters present relevant changes in model performances for different parameter values. Importantly, for almost all parameter inputs, the model shows an opposite behavior among different clusters and regions.
15   This indicates that conducting a calibration of the model against observations to determine optimal parameter values for the Central Asia domain is particularly challenging: in this case, the use of objective calibration methods is highly necessary. Finally, the sensitivity of the model to parameters perturbation for Central Asia is different than the one observed for Europe, suggesting that an RCM should be re-tuned, and its parameter uncertainty properly investigated, when setting up model-experiments to different domains of study.



## 1 Introduction

Climate models are representations of the climate system based on well-understood physics combined with simplified descriptions of sub-grid scale processes called parameterizations (Hourdin et al., 2017). These parameterizations usually depend on one or several empirical and unconfined parameters (Hourdin et al., 2017; Bellprat et al., 2012a; Tebaldi and Knutti, 2007) whose different values produce a wide spectrum of outcomes referred to as parameter uncertainty. Parameter uncertainty is important because it allows to better understand model limitations and sensitivity to different physical processes. The common approach to sample model parameter uncertainty is to use ensembles of model simulations, called Perturbed Physics Ensembles (PPEs, Murphy et al., 2007; Bellprat, 2013; Tebaldi and Knutti, 2007; Paeth, 2015).

When producing climate projections for impact studies, as much uncertainty as possible should be accounted for in order to properly drive policy-makers in their decision-making process, providing a measure of model reliability (Knutti et al., 2002, 2003; Murphy et al., 2004; Stainforth et al., 2005; Tebaldi and Knutti, 2007; Paeth et al., 2013; Bellprat, 2013; Paeth, 2015). Not properly investigating model uncertainties weakens confidence in climate projections and limits the usefulness of model outputs for adaptation strategies (Lempert et al., 2004; Foley, 2010). PPEs are of paramount importance for determining the range of model uncertainty in a probabilistic sense and assessing model reliability.

However, adequately sampling a climate model parameter space requires the performance of an extremely large number of simulations. This somehow conflicts with the need of high resolution information for impact studies and adaptation measures at a local scale. In fact, complex high resolution models have enormous computational demands, making it difficult to produce PPEs for future climate projections. Therefore, when willing to produce reliable (in a probabilistic sense) climate projections using high resolution climate models such as Regional Climate Models (RCMs), available computer resources constitute a real challenge (Paeth, 2015). A common and solid alternative practice is to use PPEs to constrain models uncertainty by selecting parameter values in a way to minimize the differences between present day observations and model results. The determined most reliable model configuration is then assumed to be the same also in the future (Hourdin et al., 2017; Bellprat et al., 2012b). This procedure is referred to as model tuning or calibration. It is important to acknowledge that calibration techniques represent only a plausible attempt to increase model reliability for climate projections, since constraining model results based on present-day skills is not a guarantee of future skill.

In recent years, the main efforts of the climate modeling community have been channeled towards developing transparent, reproducible and objective calibration methods, using well-founded mathematical and statistical frameworks (Bellprat et al., 2012b, 2016; Hourdin et al., 2017). Among others, methods based on oracle-based optimization, ensemble Kalman filters, Markov chain Monte Carlo integrations, Latin hypercubes and Bayesian stochastic inversion algorithms have been proposed and used for climate models calibration (Price et al., 2009; Beltran et al., 2006; Jackson et al., 2004; Jones et al., 2005; Annan et al., 2005; Medvigy et al., 2010; Järvinen et al., 2010; Gregoire et al., 2011; Tett et al., 2013; Schirber et al., 2013; Ollinaho et al., 2013; Williamson et al., 2013; Annan and Hargreaves, 2007). However, most of these methods cannot be directly applied to computationally costly high resolution climate models to exhaustively explore their parameter space, since typically hundreds of simulations have to be performed (Bellprat et al., 2012b; Hourdin et al., 2017). This led to the further development





of statistical surrogate models, also referred to as model emulators or meta models (O'Hagan, 2006; Bellprat et al., 2012b; Hourdin et al., 2017). These methods have the advantage to be a computationally cheap representation of the sensitivity of the climate model to the parameter space.

One of the first objective calibration methods using such a surrogate or meta model to tune an RCM is the one of Bellprat
et al. (2012a, b). Their method is mainly composed of two parts: a first one in which the model parameter uncertainty is investigated in order to determine a sub-sample of model most sensitive parameters; and a second one where a second order polynomial meta model, firstly proposed by Neelin et al. (2010), is applied to extrapolate the model behavior for all the possible values of the selected parameters and their mutual interactions. Bellprat et al. (2012b) firstly used their method for the calibration of the RCM COSMO-CLM (Rockel et al., 2008) for the Coordinated Regional Climate Downscaling Experiment
(CORDEX, Giorgi et al., 2009) European domain. The same method has successively been employed in the study of Bellprat et al. (2016) for investigating the transferability of the COSMO-CLM model configuration to other regions such as the North America CORDEX domain, and for the tuning of the same model for high resolution numerical weather predictions over Western Europe (Voudouri et al., 2017, 2018).

In this work, the results of a PPE conducted for the year 2000 with the COSMO-CLM 5.0 for the CORDEX Central Asia
domain are presented, with the main objective of investigating the model parameter uncertainty for the area and setting the basis for the application of the objective calibration method of Bellprat et al. (2012b). Central Asia is particularly important from both a climate impact and modeling perspective (Russo et al., 2019). Nonetheless, only few studies have been conducted for this area using RCMs (Ozturk et al., 2012, 2017; Russo et al., 2019) and more efforts are indeed required for better characterizing models uncertainties and limitations. Here, the results of the proposed PPE are used in order to 1) determine the COSMO-CLM
most sensitive parameters for the area, on which to apply the calibration method of Bellprat et al. (2012b) and 2) to investigate the relevance of different physical processes for different regions, assessing at the same time how much can model deficiencies be ameliorated by properly setting parameter values in each case. Additionally, the results are used to 3) address the highly debated question of transferability of RCMs model configuration to a different domain of interest (Takle et al., 2007; Jacob et al., 2007, 2012; Rockel and Geyer, 2008; Bellprat et al., 2016).

The study is structured as follows. In Section 2, the model and sensitivity simulations as well as the considered metrics are introduced. Then, the model parameter sensitivity for the entire domain is discussed in Section 3.1, while sub-regional model deficiencies are characterized in Section 3.2. A discussion on the role of different uncertainty sources on the considered metrics is presented in Section 3.3. By comparing the Central Asia setting with results obtained for the European CORDEX domain, the transferability of the model configuration between different domains is addressed in Section 3.3. Finally, the results are
summarized and conclusive remarks are presented in Section 4.





## 2    Data and Methods

### 2.1    Model and Experiments

For the simulations over Central Asia presented in this study, the regional climate model COSMO-CLM version 5.0_clm9 is used. COSMO-CLM is the climate version (Rockel et al., 2008) of the non-hydrostatic model COSMO, developed for

numerical weather predictions (Baldauf et al., 2011; Doms and Baldauf, 2013; Doms et al., 2013). It is based on the primitive thermo-hydrodynamical equations describing compressible flow in a moist atmosphere and takes into account a variety of physical processes through different parameterization schemes.

The applications of COSMO-CLM range from paleoclimate (Russo and Cubasch, 2016; Fallah et al., 2016; Prömmel et al., 2013) to future projections (Panitz et al., 2014; Bucchignani et al., 2016, 2014; Fischer et al., 2013; Dobler and Ahrens, 2011;

Wang et al., 2013; Keuler et al., 2016; Sørland et al., 2018), and span a large variety of spatial resolutions, from meso- to convection-permitting scales (Fosser et al., 2015; Knote et al., 2010; Brisson et al., 2015; Tölle et al., 2014; Ban et al., 2015).

The model is used in this study at a spatial resolution of 0.22°, following the framework of the new CORDEX Coordinated Output for Regional Evaluations (CORDEX-CORE) initiative (Gutowski Jr et al., 2016, see web: https://www.cordex.org/experiment-guidelines/cordex-core/).

The simulations used for characterizing the model parameters uncertainty are 1-year long and are run from an equilibrium state obtained from a 10-year long simulation over the period 1991-2000. This simulation represents the reference simulation for this study. Its configuration is derived from Russo et al. (2019) and uses a representation of vegetation albedo taking into account forest fraction and soil heat conductivity accounting for soil ice/moisture ratio. Table 1 provides a summary of the reference simulation setup. A more detailed description of the model domain (showed in Fig. 1) and reference configuration is

presented in Russo et al. (2019). Bellprat et al. (2012a) demonstrated that, for some pre-defined metrics, COSMO-CLM results converge already after one year. Therefore, for the purpose of determining the model most sensitive parameters, although perturbed physics experiments should ideally cover 3-to-5 years, 1-year long simulations could be enough, especially when computational resources are limited. Here, the year 2000 has been selected, since it can be considered normal in terms of monthly values of the investigated variables.

The tested parameters inputs are selected from a plausible range derived from Bellprat et al. (2012a), with a minimum, a maximum and different intermediate values, depending on the parameter. A list with all the tested values is presented in Table 2, for a total number of ninety-two simulations.

For the analysis, an estimate of the internal variability of the model is needed. Thus, an ensemble of five simulations covering the period 1991-2005 but with different initial conditions (starting date shifted by ±1 and ±3 months), performed by Russo

et al. (2019), is additionally considered.

To investigate the model transferability to a different domain, the same 1-year long perturbed simulations are performed for four parameters for the European CORDEX domain, yielding an additional fifteen simulations. An ensemble of five 15-year long simulations with different initial conditions is also performed for Europe in order to estimate the model internal variability for the region.



All the presented simulations are driven by NCEP version 2 (NCEP2) reanalysis data (Kanamitsu et al., 2002). NCEP2 data have a temporal resolution of 6 hours and a spectral resolution of T62 ($\sim 1.9°$). Normally, ERAInterim reanalysis data are used to drive RCMs evaluation and calibration experiments. Conversely, NCEP2 data are employed in this study, with the specific purpose of reproducing the spatial resolution jump present when using the Global Circulation Models (GCMs)

normally employed in CORDEX simulations ($\sim 200$ km, Russo et al., 2019).

## 2.2   Observations

The presented analyses focus on three variables: near surface temperature (T2M), daily precipitation (PRE) and total cloud cover (CLCT). While T2M and PRE are higly relevant for climate impact studies, the third one is used to evaluate models ability in simulating radiative processes (Bellprat et al., 2012a, b, 2016).

The range of three different observational data sets is considered for each of the variables to represent observational uncertainties (Collins et al., 2013; Gómez-Navarro et al., 2012; Bellprat et al., 2012a, b; Flaounas et al., 2012; Lange et al., 2015; Zhou et al., 2016; Solman et al., 2013; Russo et al., 2019):

- For temperature, information is retrieved from the CRU TS4.1 observational data set (Harris and Jones, 2017), from the University of Delaware (UDEL) gridded data set (Willmott, 2000), provided by the NOAA/OAR/ESRL PSD, Boulder,

Colorado, USA, from their Web site at https://www.esrl.noaa.gov/psd/, and from the Modern-Era Retrospective analysis for Research and Applications, version 2 (MERRA2) (Gelaro et al., 2017).

- Information on precipitation is retrieved from the CRU and the UDEL data sets, as well as from the Global Precipitation Climatology Centre dataset (GPCC) (Becker et al., 2011).

- For total cloud cover, again the CRU data set is used. Additionally, the cloud data sets extracted from the National

Oceanic and Atmospheric Administration (NOAA) High Resolution Infrared Radiation Sounder (HIRS) (Wylie et al., 2005) and the International Satellite Cloud Climatology Project (ISCCP) gridded dataset (Zhang et al. 2004) are used, similarly to Bellprat et al. (2012a, b).

All data sets present a spatial resolution of $0.5°$, except the HIRS and the ISCCP, having both a spatial resolution of $1°$. In the latter case the data are interpolated on the CRU grid by means of a conservative remapping method prior to the analyses.

The considered observational data sets and the corresponding variables for which they are used are reported in Table 3.

## 2.3   Analysis Methods and Evaluation Metrics

The analyses are conducted on the regional means of monthly values of the considered variables for different regions characterized by differing climate conditions. After averaging, the model residuals with respect to observations become quasi-Gaussian, allowing the use of normal estimators of model disagreement (Von Storch and Zwiers, 2001; Bellprat et al., 2012a).

A k-means clustering technique (Steinhaus, 1956; Ball and Hall Dj, 1965; MacQueen et al., 1967; Lloyd, 1982; Jain, 2010; Russo et al., 2019) applied onto quantile-normalized (q-normalized) monthly climatologies of the investigated variables is used





to decompose the domain into a set of sub-regions with different climate conditions. K-means allows the separation of similar data into groups, using the concept of Euclidean distance from the centroids of a pre-determined group of clusters. Following several tests and the results of other studies (Mannig et al., 2013; Russo et al., 2019) a total number of eleven clusters have been selected for the Central Asia domain. As input for the clustering procedure, q-normalized values of monthly climatologies of

T2M and CLCT derived from the CRU data set and PRE values derived from the GPCC are used. The results of the k-means clustering are shown in Fig. 2.

The metrics used for investigating the COSMO-CLM parameters uncertainty is the Performance Index (PI) presented in Bellprat et al. (2012a) and derived from the Climate Performance Index (CPI) of Murphy et al. (2004). PI represents a normalized multivariate root-mean-square error (RMSE), weighted over different sources of uncertainties and averaged over the

model variables, the considered regions and the months of a selected year:

$$ \mathrm{PI} = \frac{1}{VRT} \sum_v^V \sum_r^R \sum_t^T \frac{\sqrt{(m_{v,r,t} - o_{v,r,t})^2}}{\sigma_{o_{v,r,t}} + \sigma_{iv_{v,r,t}} + \sigma_{\epsilon_{v,r,t}}} \tag{1} $$

where $V = 3$ represents the number of variables considered, $R = 11$ is the number of the domain sub-regions and $T = 12$ is the number of months of the given year. The terms *m* and *o* represent the model and the observational monthly means calculated for each variable, month and region. $\sigma_o$ is the monthly standard deviation of the interannual variations calculated from the

observations over the period 1996-2005; $\sigma_{iv}$ is the monthly standard deviation of the internal variability of the regional model for the same period; $\sigma_\epsilon$ is the monthly standard deviation of the observational error derived from different reference datasets, for the selected year.

PI represents an objective measure of model reliability, where higher (lower) values indicate bad (good) performances. In order to make inferences about the sensitivity of model parameters, Bellprat et al. (2012a, b) used the PI to define a positive

Performance Score (PS), that can be interpreted as an approximation of the likelihood that the residuals come from a distribution with zero mean and variance given by $\sigma_o$, $\sigma_{iv}$ and $\sigma_\epsilon$:

$$ \mathrm{PS} = e^{(-0.5\mathrm{PI}^2)} \tag{2} $$

Basically, PI allows to quantify model parameter uncertainty, while PS is used as an estimate of the model sensitivity to each single tested parameter.

In this study, first PI and PS are calculated for the three considered variables together. Then, given the assumption that changes in PS are expected to be smooth (Bellprat et al., 2012a; Neelin et al., 2010), a quadratic regression is fitted to the obtained values of PS for each parameter, representing an estimate of model sensitivity for that specific parameter. Successively, the same analyses are repeated for each variable separately, taking into account the fact that the obtained PS values might be due to a compensation effect of the results for single variables. This will contribute to discriminate the model most sensitive

parameters for the region.





In a successive step, model parameter uncertainties for different areas of the domain are investigated. For this purpose PI is calculated separately for each variable and sub-region. Then, the variable and region dependent PI is expressed with respect to the one of the reference simulation using a Skill Score (SS) defined as:

$$SS = (1 - \frac{PI_{exp}}{PI_{ref}}) \tag{3}$$

Positive (negative) SS values indicate an improvement (worsening) of the considered experiment over the reference simulation, in terms of the proposed metrics PI.

The range of different errors and their effects on the considered metrics will be additionally investigated to support the presented analyses.

Finally, for the comparison of the model results obtained for Central Asia with the ones for Europe, the same PS metrics,
calculated for a sub-set of selected parameters over the entire domain, will be considered.

## 3 Results

### 3.1 Sensitivity of the Model to Parameters Perturbation for the Entire Domain

First, the performance score (PS) for each parameter, when considering T2M, PRE and CLCT together, is investigated (Fig. 3). It is clearly seen that model performances are sensitive to only a restricted set of parameters, which is in agreement with
the findings of Bellprat et al. (2012a). The parameters that have the largest impact on PS are **e_surf**, representing the exponent to get the effective surface area used in the land-surface scheme, and **qi0**, being the parameter for the cloud ice treshold for autoconversion used in the microphysics parametrization scheme. Other parameters, which have some considerable impact on PS, are **d_mom**, the factor for turbulent momentum dissipation, **v0snow**, controlling the fall velocity of snow, **radfac**, which represents the fraction of clouds water/ice used in the radiation scheme, **tkhmin**, the minimum value for the turbulence heat
diffusion coefficient, and **rlam_heat**, the scaling factor of the laminar boundary layer for heat. Thus, for each parametrization scheme, excluding convection, there is at-least one or two parameters that shows the potential to sensibly improve model performances when an optimal value is set. For some of the parameters such as **c_diff**, the factor for turbulent diffusion in the turbulent kinetic Energy (TKE) scheme, and **z0m_dia**, representing the roughness length of a typical synoptic station used for the interpolation of values of the 10-m wind, strong changes in PS are evident in Fig. 3. However, in these cases the model
performs similarly for all tested inputs, suggesting that the evinced sensitivity is an artificial result of the quadratic interpolation. Changes in PS are also evident for **soilhyd**, a multiplying factor for soil hydraulic conductivity and diffusivity, **fac_rootdp2**, an uniform factor for the root depth field, **tur_len**, defining the maximal turbulent length scale, **uc1**, used for computing the amount of cloud cover in saturated conditions, and **q_crit**, representing the critical value for normalized over-saturation. For all other parameters, variations of PS are considerably small or zero.
When investigating how PS depends on each variable (Fig. 4), similar results are obtained for all the parameters. Largest variations in PS are evident, for each of the variables, for **e_surf** and **qi0**. Remarkable changes in PS for T2M are also identified




for **tkhmin**. In this case, also **c_diff** shows significant changes but, as already stated above, the PS seems to be at its maximum for the parameter values lower and higher limits, suggesting that any parameter input in this range will not produce an improvement in temperatures. For PRE, more pronounced variations of PS are also found for **d_mom**, **v0snow** and **rlam_heat**. Finally, for CLCT, considerable changes in PS are also evident for **tkhmin**, showing an opposite behavior with respect to the one of

temperatures. Other parameters are characterized by particularly small variations in the PS calculated for single variables, but these changes are coherent among all the different variables and translate into slightly larger changes in the PS calculated over the three variables together. This is the case of **radfac**, **soilhyd**, **tur_len**, **rat_lam**, **uc1** (Fig. 3). In all other cases, variations of the PS calculated for different variables compensate each other, leading to really small or zero changes in the total PS. In general, it is important to notice that the values of PS are lower for PRE than for the other two variables.

Based on these results, the nine most sensitive model parameters for the region, highlighted in blue in Tab. 3, are identified. A maximum of two parameters are selected for each of the model physical schemes (turbulence, surface, soil, radiation, and grid scale clouds precipitation), excluding convection, for which the only tested parameter **entr_sc**, representing the mean entrainment rate for shallow convection, leads to really similar results for all the considered inputs. Thus, to properly set an optimal model configuration for the Central Asia domain, taking into account computational costs constraints, the selected nine

parameters are recommended to conduct the objective calibration procedure of Bellprat et al. (2012b).

### 3.2   Model Behavior for Different Sub-regions

Once the model most sensitive parameters for the area are identified, a more detailed analysis of simulation results for each variable and sub-region is performed. The aim is to investigate the model parameter uncertainties for regions characterized by different climate conditions, determining the most relevant processes in the model formulation in each case and assessing to

which degree it is possible to reduce biases in the considered variables, by properly setting parameter values.

Figures 5, 6 and 7 show (top panel) the PI calculated for the reference simulation (simulation with default values, highlighted in red in Table 2) for each of the sub-regions for T2M, PRE and CLCT, respectively, and the changes in the SS, for each cluster and performed experiment (lower panel of Fig. 5, Fig. 6 and Fig. 7). The figures illustrate the magnitude of model deficiencies for the reference simulation, allowing to evaluate at the same time parameters sensitivity for each sub-region and variable. The

high PI values evident for PRE, confirm that the model performances are particularly poor with respect to this variable.

Fig. 5 (upper panel) shows that the largest mismatches between the reference simulation and observations, in terms of T2M, are found over the Tibetan Plateau (TIB, Fig. 2). This particularly strong cold bias in temperature over the Tibetan Plateau, found for all seasons, is a common feature of several RCMs, as discussed in Russo et al. (2019). Some studies highlighted the importance of a better representation of surface features and processes for the Tibetan Plateau, characterized by particularly

complex topography (Meng et al., 2018; Zhuo et al., 2016). Here, the results indicate that, for COSMO-CLM, parameters characteristic of surface parameterizations play only a secondary role for the simulation of T2M over the region, with pronounced changes in model performances evident only for few parameters such as **e_surf** and **pat_len**, with the latter expressing the length scale of subscale surface patterns over land. The largest improvements in the COSMO-CLM simulation of T2M over the Tibetan region are obtained for the parameter **qi0**, characteristic of the cloud grid-scale condensation (microphysics) physi-





cal scheme. Additionally, large variations in SS are also obtained for the same region for the parameter **tkhmin**. Another region where microphysics parameterizations and the characterization of cloud grid-scale condensation lead to an improvement in the simulated T2M is the Northern India Monsoon (IMO, Fig. 2) area. Particularly high values of PI for T2M are also evident in the upper panel of Fig. 5, for the Arabian Peninsula and the Southern Iran region (DHS, Fig. 2): no clear improvements in
model results relatively to this variable can be obtained for this region by perturbing parameter values (bottom panel, Fig. 5). This suggests that possible model deficiencies in this case are likely related to the model formulation itself. This seems to be also the case for the northern part of the domain, corresponding to Western Siberia (SAR, CSA and DSS, Fig. 2). Russo et al. (2019) showed that the COSMO-CLM presents particularly poor performances in the simulation of temperatures over Western Siberia, specifically for winter, with warm biases deriving from the comparison against observations exceeding 15°C over some
points. The presented analyses show that changes in PI for T2M are quite consistent among the considered Western Siberia sub-domains. In this case, improvements in model performance for the simulation of T2M in terms of PI are limited to a few parameters. When the Western Siberia sub-regions are considered together, main improvements are obtained for **rlam_heat**, **radfac**, **hincrad**, **c_land** and **e_surf**, indicating the importance of surface and soil features and processes related to heat fluxes for the simulation of temperature over the area. Nevertheless, these changes do not seem to be large enough to significantly
reduce the T2M model bias over Western Siberia, pointing at a structural problem in the model formulation. For all other regions, there seems to be some potential to improve model performances in simulating T2M, by properly choosing parameters values. Largest positive values of SS are obtained for the Northern Black Sea (WSC) region and the transition zone between the Northern India Monsoon and the Tibetan region (MTT). For T2M no parameter value seems to lead to an univocal positive model response over all the different clusters together. For most of the regions, properly calibrating the parameters **qi0**, **tkhmin**
and **e_surf** is particularly important in order to avoid significantly poor model performances in the simulation of T2M.

    For PRE (Fig. 6), the amplitude of changes in the calculated SS are smaller than for T2M and CLCT. The region for which an important improvement in model performances is possible, for almost all considered parameters, is the region covering the desert areas of Mongolia and Northwestern China (DCW, Fig. 2). Large improvements in the simulation of PRE are also possible for the region covering the northern part of Iran and Turkey (SDT, Fig. 2) where moisture is mainly advected by
westerlies from the Mediterranean Sea (Fernández et al., 2003; Fallah et al., 2015). In this case, the model seems particularly sensitive to changes in the parameter **c_sea**, describing the surface area density of the waves over sea, used for the calculation of roughness length. The roughness of the sea surface steers the exchange of momentum, moisture and heat between ocean and atmosphere (Carlsson et al., 2010; Vickers and Mahrt, 2010). Thévenot et al. (2016) demonstrated that an increase in sea surface roughness may generate higher momentum fluxes, impacting low level atmospheric dynamics, particularly affecting wind
speeds, and that a proper representation of the sea surface roughness may lead to a better localization of heavy precipitation. For the northern part of Iran and Turkey, the other parameter for which the model is most sensitive to the simulation of PRE is **rat_sea**, the ratio of laminar scaling factors for heat over sea and land, confirming again the importance of the representation of ocean-atmosphere interactions for the simulation of precipitation over the region. Finally, significant improvements in simulated PRE are obtained for the Tibetan Plateau (TIB, Fig. 2), for the parameter **qi0**, affecting clouds microphysics,
as previously described. For all other regions, changes in model performance for the simulation of PRE are not particularly





remarkable, making any assumption on parameters selection almost equivalent. Only the parameter **d_mom** seems to be able to produce small, but positive, improvements with respect to the reference simulation, for all the clusters.

For CLCT, changes in the SS are significant only for a specific sub-set of parameters (Fig. 7). In particular, for **qi0**, mainly negative values are obtained for all the regions. Considering the opposite positive effect of this parameter inputs on T2M for
some regions, properly selecting a value for the parameter **qi0** becomes of fundamental importance for COSMO-CLM simulations over the CORDEX Central Asia domain, with even tiny changes in the parameter input that could have dramatic effects on model performances for different variables. The same holds true for the parameter **e_surf**. Important SS variations for CLCT are evident, in the bottom panel of Fig. 7, for the north-western areas of the domain (SAR, CSA, WSC and STE, Fig. 2), for the parameter **tkhmin**, characterizing heat turbulent diffusion. These regions are characterized by particularly stable stratified
atmospheric conditions. For these, the model has already proved to be highly sensitive to **tkhmin** (Cerenzia et al., 2014; Buzzi et al., 2011), producing excessive mixing during periods with highly stable stratification and a consequent overestimation of temperatures. Basically, higher values of **tkhmin** produce exaggerated mixing, leading to more cloud formation, more similarly to observations, that otherwise the model is not able to reproduce. However, these improvements are only inherent to the simulated CLCT, and the same processes lead to higher T2M with a consequent worsening of the model results over the same
region. This represents a case where the model generates better results for the wrong reason. In this case, parameter inputs must be carefully selected, and the application of an objective calibration method becomes indispensable. Another parameter that presents the potential to sensibly improve model performances for CLCT, for several domain sub-regions, is **uc1**. This parameter shows an opposite behavior between southern (DHS, SDT, MTT and IMO, Fig. 2) and northern regions.

In general, the results from Figs. 5, 6 and 7 indicate that the computed SS does not exhibit a similar and coherent behavior
over all sub-domains for the different parameters tested. This suggests that even if properly setting COSMO-CLM parameter values for Central Asia could lead to a general improvement of model performances for the entire domain, this improvement would not be absolute: it will likely be the result of relatively poorer performances over specific areas compensated by larger improvements over other regions.

### 3.3 Considerations on Different Uncertainty Sources

To better understand the role of different uncertainty sources on the calculation of PI, here a more detailed analysis of the considered errors is presented. Fig. 8 shows the values of the different uncertainty terms considered in the calculation of PI. For T2M and PRE the highest uncertainties are obtained from the observational interannual variability ($\sigma_o$), for almost all months and clusters. In the first case the highest uncertainties characterize winter months, especially over Western Siberia (SAR, CSA and DSS, Fig.2) and the Steppe region East of the Caspian Sea (STE, Fig.2). In the second, highest uncertainties
are evident for summer months over the monsoon areas (MTT and IMO, Fig.2). Conversely, for CLCT, the largest contribution to the sum of the uncertainties is given by the mean differences in the considered observational data sets ($\sigma_\epsilon$) for all months and regions.

Fig. 9 shows the effect of the sum of the different uncertainty terms on the calculation of PI for each month and cluster, relatively to the corresponding model bias against observations. The presented values are obtained by standardizing the value





of the uncertainty/bias ratios, for all variables, with respect to their absolute minimum and maximum values. From this figure it is possible to see that for the calculation of PI, uncertainties have a greater weight for T2M and CLCT than for PRE. This suggests that the lower PS values obtained for PRE are mainly due to particularly large model biases in this case, rather than to uncertainties in the observational datasets.

### 3.4 Transferability of the Model Configuration

In order to test whether the COSMO-CLM responds similarly to parameters perturbation for different domains, the same PS analyses are conducted on a sub-set of model parameters for the European CORDEX domain. The same 1-year long simulations for the year 2000 are conducted for the parameters **e_surf**, **rlam_heat**, **rat_sea** and **entr_sc**, using the same default configuration of the Central Asia simulations. The same k-means clustering technique is used for dividing the European domain into sub-regions characterized by different climatic conditions. For having approximately an equal ratio between the total number of points and clusters as for Central Asia, six clusters are selected. Results of the clustering for Europe are shown in Fig. 10.

The PS calculated for the considered parameter values for Europe and Central Asia is presented in Fig. 11, respectively in the left and in the right columns. The results show that the model has the same sensitivity over Europe and Central Asia for **e_ surf**, with worse performances for higher parameter values. Conversely, a different behavior is obtained in the two cases for the other parameters. In particular, the model shows for Europe a decreasing response to an increase in parameter values for **rlam_heat**, differently to the Central Asia case. At the same time, for Europe, **rat_sea** and **entr_sc** present appreciable changes, in particular the first one, conversely to the Central Asia example. One important aspect to be mentioned is the fact that the values of PS calculated for Europe are higher than the ones for Central Asia differ. An additional analysis of the uncertainty for Europe, indicates that this difference is dictated by larger model biases against observations for Central Asia, since the ranges of the uncertainties are comparable in the two cases (see supplements). Even though the presented analyses are conducted on a sub-sample of parameters, the results allow to affirm that the model responds differently to different parameter values over Europe and Central Asia, two regions characterized by various climate conditions. This is in contrast with the recent findings of Bellprat et al. (2016), asserting that uncertainties in the model physics are common among different regions. Even though additional research on other regions is required, the presented results suggest that model calibration remains a necessary condition prior to the application of COSMO-CLM to different domains of study.

## 4 Conclusions

In this paper the parameter space of the Regional Climate Model (RCM) COSMO-CLM is investigated for the CORDEX Central Asia domain, using a Perturbed Physics Ensemble (PPE) obtained by performing 1-year long simulations with different parameter values. The results of these simulations are compared against observations, using the performance metrics introduced in Bellprat et al. (2012a). The main goal of the paper is to characterize model parameter uncertainty, determining the most sensitive parameters for the region, on which to apply the objective calibration method of Bellprat et al. (2012b). Moreover,





the presented experiments are used to investigate the effect of several parameters on the simulation of the considered variables for sub-regions with different climate conditions, assessing by which degree it is possible to improve model performances by properly selecting parameter values in each case. Finally, the paper explores the possibility of transferring an RCM model setup used for one region, to a different domain of study.

The model is particularly sensitive to a sub-set of all the tested parameters. The parameters with the largest effect on model performances are **qi0**, the cloud ice threshold for autoconversion, and **e_surf**, the exponent to get the effective surface area. Another particularly important parameter for the area and all considered variables is **rlam_heat**, the scaling factor of the laminar boundary layer for heat. In addition to these three, six other most-sensitive parameters are individuated: **d_mom**, the factor for turbulent momentum dissipation, **v0snow**, controlling the fall velocity of snow, **radfac**, which represents the fraction

of clouds water/ice used in the radiation scheme, **tkhmin**, the minimum value for the turbulence heat diffusion coefficient, **soilhyd**, a multiplying factor for soil hydraulic conductivity and diffusivity and **uc1**, the parameter for computing the amount of cloud cover in saturated conditions.

    In general, the presented results show that an overall improvement of model performances relatively to the selected variables seems possible, by properly selecting parameter values. Nevertheless, this improvement would not be coherent among all

the Central Asia domain sub-regions, but the result of some compensating effect. The model response to parameter values perturbation is characterized by contrasting results in the different cases. The most important example in this sense is the one of **qi0**, producing large contrasting changes in model performances over different regions and variables. It is of crucial importance to determine a proper input for **qi0** for the region, since even a small perturbation of its value could have a tremendous effect on model results, with an improvement in the performances over one region and variable, and an opposite response for others.

The same is particularly true also for the parameter **e_surf**.

    Parameters related to soil and land-surface atmosphere interactions representation, such as **rlam_heat** and **soilhyd**, are notably relevant for the simulation of near surface temperature over a large part of the domain sub-regions, in particular over Western Siberia and the area North of the Black Sea. Nevertheless, the same parameters do not have the same influence on the simulation of precipitation and total cloud cover for the majority of the sub-domains. Parameters used in the turbulence

parameterization scheme, such as **tkhmin**, have an important impact on many regions, in particular for near surface temperature and cloud cover, for areas characterized by complex topography and the ones with stable vertical stratification. In the latter case, **tkhmin** produces opposite results for the considered variables, confirming an already known model structural problem related to the production of excessive mixing over these regions. Among the parameters employed in the radiation processes representation, **uc1** shows a strong sensitivity, in particular for total cloud cover, over all the domain sub-regions. Over some

sub-regions ( e.g. Turkey and the northern part of Iran ) parameters related to ocean-surface processes, such as **rat_sea** and **c_sea**, have a relevant effect on the simulation of precipitation. For some regions, such as Western Siberia, even though changes in model results are possible by perturbing parameter values, they do not seem to be large enough in order to sensibly improve model biases. In this case, the reason for the biases is most likely related to some structural error in the model formulation.

    For the calculation of the considered metrics, a larger role is played by the uncertainties in near surface temperature and

cloud cover, than the ones in precipitation. For cloud cover, the contribution of the observational uncertainties is larger than the




one arising from the observational interannual and model internal variability, more important for the other two variables. The bias between model and observational data sets is particularly large for precipitation, leading to lower values of the considered Performance Score (PS) with respect to the other two cases.

Finally, the model parameters sensitivity for the CORDEX Central Asia domain does not coincide, for a selected sub-sample
of parameters, with the one evinced for the corresponding European domain, characterized by different climatic conditions. Even though additional research on other regions is required, the presented results suggest that model calibration remains a necessary condition prior to the application of COSMO-CLM to different domains of study, contrary to evidence from recent studies using the same model. Our results suggest that a regional climate model should be re-tuned when setting up model-experiments to a non-native domain.

*Code and data availability.*

Simulations configuration files can be downloaded from:
https://doi.org/10.5281/zenodo.3523177
All the data on which the presented analyses are conducted, together with the restart files of the reference simulations used to drive the sensitivity experiments for the two domains, are available at the following link:
https://doi.org/10.5281/zenodo.3523243
A complete documentation of the COSMO-Model is permanently available at the following link:
https://www.dwd.de/EN/ourservices/cosmo_documentation/cosmo_documentation.html
The COSMO-CLM model is completely free of charge for all research applications. The version of the COSMO-CLM model used in this study can be downloaded from the following website:
https://redc.clm-community.eu/projects/cclm-sp/wiki/Downloads.
Access is license-restricted (http://www.cosmo-model.org/content/consortium/licencing.htm) and for the download the user needs to become a member of the CLM-Community, or the respective institute needs to hold an institutional license.

*Author contributions.* The simulations of this research were performed by ER. All the authors contributed to the discussion of the results, with ER and SS playing the major role. The paper structure as well as most of the presented experiments were designed by ER, with the
support of SS and CR. All authors gave a substantial contribute to the revision of the text and to the formatting of the paper.

*Competing interests.* No competing interests are present in the paper.



*Acknowledgements.* This study was funded by the Federal Ministry of Education and Research of Germany (BMBF) as part of the CAME II project (Central Asia: Climatic Tipping Points & Their Consequences), project number 03G0863G.

The computational resources necessary for conducting the experiments presented in this research were made available by the German Climate Computing Center (DKRZ).

5   The authors are also particularly grateful to the COSMO and the CLM-Community for all their efforts in developing the COSMO-CLM model and making its code available.

Finally, a special acknowledgment goes to Uli Schättler, Daniel Rieger and their collaborators from the German Weather Service (DWD) for making the COSMO-Model documentation permanently available.



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



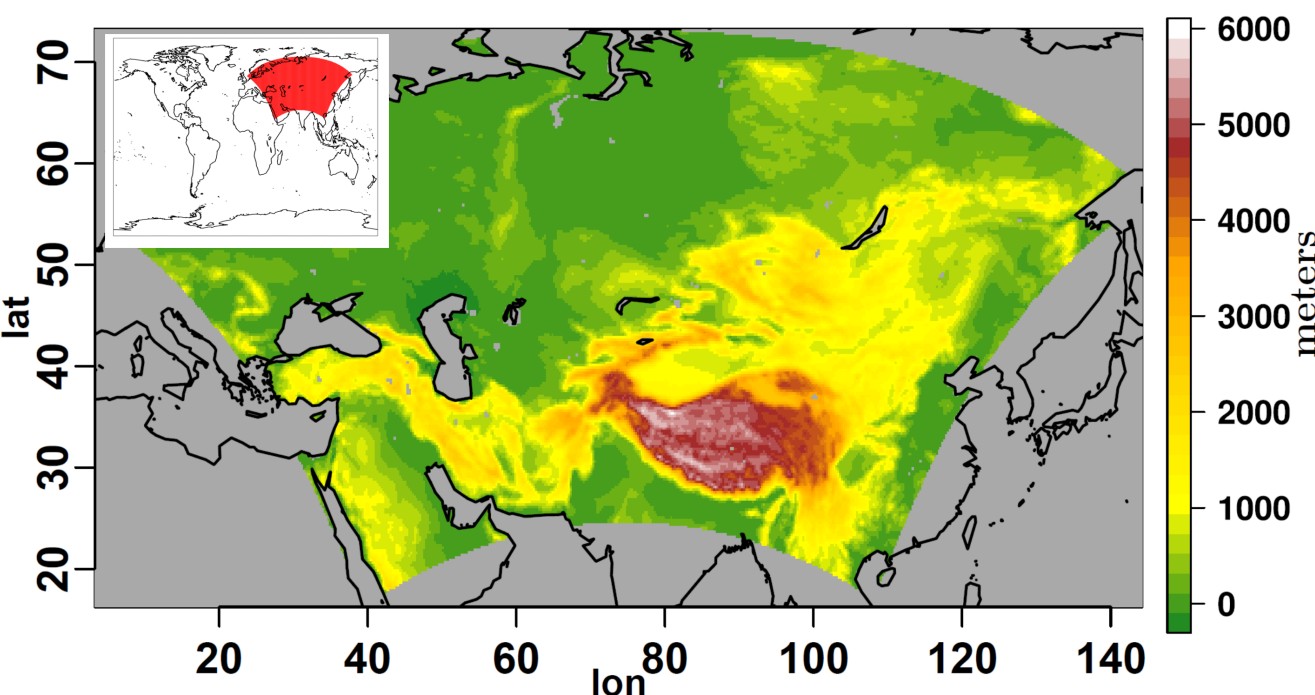

**Figure 1.** Location and Orography map of the Central Asia domain, at a spatial resolution of 0.22°. Orographic data are derived from the Global Land One kilometre Base Elevation (GLOBE , Hastings et al., 1999) data set.

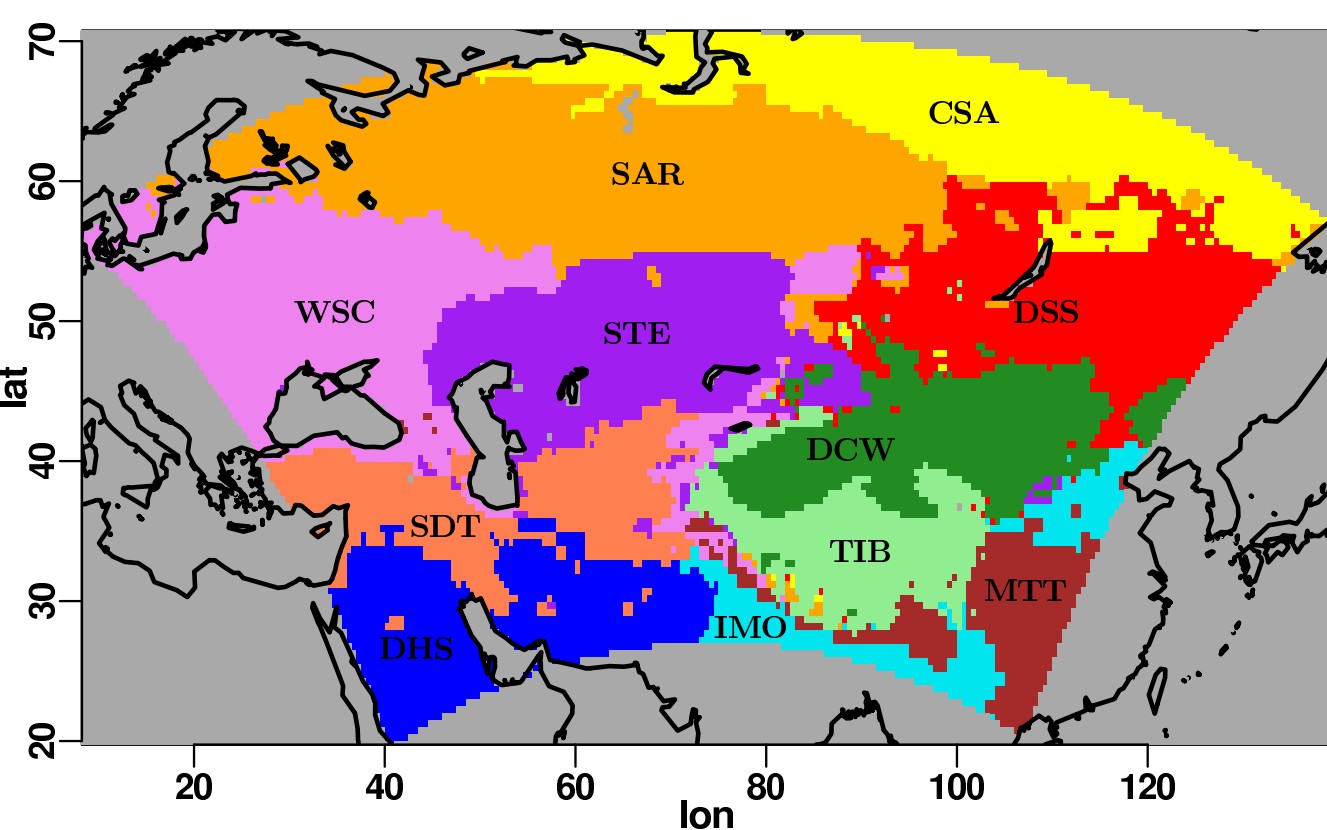

**Figure 2.** Map of the eleven sub-regions obtained through k-means clustering of q-normalized monthly climatologies of the three considered variables over the period 1996-2005.



**Figure 3.** PS calculated for near surface temperature (T2M), daily precipitation (PRE) and total cloud cover (CLCT) together, for all the different tested parameter values, over the entire Central Asia domain.





**Figure 4.** PS calculated separately for near surface temperature (T2M, solid line), daily precipitation (PRE, dashed line) and total cloud cover (CLCT, dotted line), for all the different tested parameter values, over the entire domain.



**Figure 5.** *Top*: PI calculated for near surface temperature (T2M) for the reference experiment with default parameter values, over each of the eleven Central Asia sub-regions. *Bottom*: Changes in the SS of each performed experiment, calculated with respect to the default simulation, for near surface temperature (T2M): green (violet) values indicate a better (worse) agreement with observations with respect to the default simulation. The experiments for each parameter are enumerated in an increasing order, according to its tested values, from the lowest to the highest.



**Figure 6.** *Top*: PI calculated for precipitation (PRE) for the reference experiment with default parameter values, over each of the eleven Central Asia sub-regions. *Bottom*: Changes in the SS of each performed experiment, calculated with respect to the default simulation, for precipitation (PRE): green (violet) values indicate a better (worse) agreement with observations with respect to the default simulation. The experiments for each parameter are enumerated in an increasing order, according to its tested values, from the lowest to the highest.

**Figure 7.** *Top*: PI calculated for total cloud cover (CLCT) for the reference experiment with default parameter values, over each of the eleven Central Asia sub-regions. *Bottom*: Changes in the SS of each performed experiment, calculated with respect to the default simulation, for total cloud cover (CLCT): green (violet) values indicate a better (worse) agreement with observations with respect to the default simulation. The experiments for each parameter are enumerated in an increasing order, according to its tested values, from the lowest to the highest.



**Figure 8.** Values of different uncertainty terms for each cluster and month, used in the PI calculations. From top to bottom the different variables are considered, in the following order: near surface temperature (T2M), daily precipitation (PRE) and total cloud cover (CLCT). The left column represents the uncertainties of the observation interannual variability ($\sigma_O$) calculated over the period 1996-2005. In the central column the values of the model internal variability ($\sigma_{iv}$), calculated over the same period, are shown. In the right column the errors calculated over the different observational data sets, for the selected year 2000, are illustrated ($\sigma_\epsilon$).







**Figure 9.** Effects of the sum of the different uncertainty sources in the calculation of PI for each cluster and considered month, with respect to the reference model bias against observations. The error terms are firstly divided by the bias of the reference simulation for each sub-region and month. Then, the values obtained for all variables are standardized between a minimum and a maximum.



**Figure 10.** Map of the six sub-regions obtained for Europe through k-means clustering of the q-normalized monthly climatologies of the considered variables, calculated over the period 1996-2005.



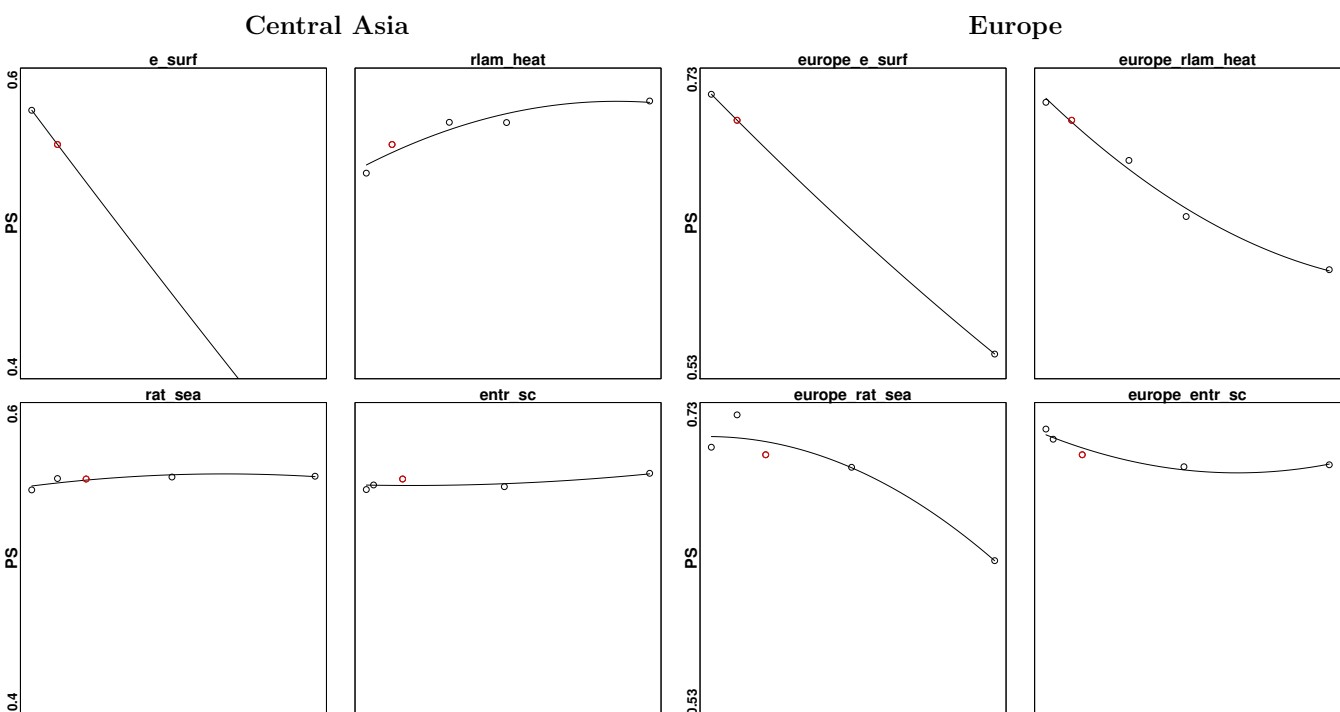

**Figure 11.** PS values calculated for Central Asia (*left*) and Europe (*right*), for different values of the parameters **e_surf**, **rlam_heat**, **rat_sea** and **entr_sc**. The values of the parameters are the same in the two cases. Red dots represent the considered parameter values for the default simulation.



**Table 1.** General description of model setup of the reference simulation

| | |
|---|---|
| **Spatial Resolution** | $\approx 0.22°$ |
| **Timestep** | 150s |
| **Convection** | Tiedke |
| **Time Integration** | Runge-Kutta, |
| **Lateral Relaxation Layer** | 250 km |
| **Soil Model** | TERRA-ML SVAT |
| **Aerosol** | Tegen (Tegen et al., 1997) |
| **Albedo** | Vegetation albedo function of forest fraction |
| **Rayleigh Damping Layer (rdheight)** | 18 km |
| **Soil Active Layers** | 9 |
| **Active Soil Depth** | 5.74 m |
| **Atmospheric Vertical Layers** | 45 |





**Table 2.** List of model parameters and corresponding ranges of investigated input values. The parameter values of the default model configuration are reported in red. The finally selected COSMO-CLM most-sensitive parameters for the region are highlighted in blue.

| Parameter | Description | Values |
|---|---|---|
| **Turbulence** | | |
| **tkhmin** | minimal diffusion coefficients for heat | (0,0.4,1,2) |
| **tkmmin** | minimal diffusion coefficients for momentum | (0,0.4,1,2) |
| **tur_len** | maximal turbulent length scale | (100,500,1000) |
| **d_heat** | factor for turbulent heat dissipation | (12,10.1,15) |
| **d_mom** | factor for turbulent momentum dissipation | (12,15,16.6) |
| **c_diff** | factor for turbulent diffusion of TKE | (0.01,0.2,10) |
| **q_crit** | critical value for normalized over-saturation | (1,4,7,10) |
| **clc_diag** | cloud cover at saturation in statistical cloud diagnostic | (0.2,0.5,0.8) |
| **Land Surface** | | |
| **rlam_heat** | scaling factor of the laminar boudary layer for heat | (0.1,1,3,5,10) |
| **rat_sea** | ratio of laminar scaling factors for heat over sea and land | (1,10,20,50,100) |
| **rat_can** | ratio of canopy height over z0m | (0,1,10) |
| **rat_lam** | ratio of laminar scaling factors for vapour and heat | (0.1,1,10) |
| **c_sea** | surface area density of the waves over sea [1/m] | (1,1.5,5,10) |
| **c_lnd** | surface area density of the roughness elements over land | (1,2,10) |
| **z0m_dia** | roughness length of a typical synoptic station | (0.001,0.2,10) |
| **pat_len** | length scale of subscale surface patterns over land | (10,100,500,1000) |
| **e_surf** | exponent to get the effective surface area | (0.1,1,10) |
| **Convection** | | |
| **entr_sc** | mean entrainment rate for shallow convection | (5e-5, 1e-4, 3e-4,1e-3, 2e-3) |
| **Microphysics** | | |
| **cloud_num** | cloud droplet number concentration | (5e+7,5e+8,1e+9) |
| **qi0** | cloud ice threshold for autoconversion | (0,0.00001,0.0001,0.001,0.01) |
| **v0snow** | factor for fall velocity of snow | (10,15,25) |
| **Radiation** | | |
| **uc1** | parameter for computing amount of cloud cover in saturated conditions | (0.2,0.5,0.625,0.8) |
| **hincrad** | increment for running the radiation in hours | (0.5,0.75,1) |
| **radfac** | fraction of cloud water/ice used in radiation scheme | (0.3,0.5,0.9) |
| **Soil** | | |
| **soilhyd** | multipl. factor for hydraulic conductivity and diffusivity | (1,1.62,6) |
| **fac_rootdp2** | Uniform factor for the root depth field | (0.5,1,1.5) |





**Table 3.** List of observational and reanalysis data sets employed for the evaluation of model results.

| Observational Dataset | Variables |
| --- | --- |
| **CRU TS4.1** | T2M, PRE, CLCT |
| **UDEL** | T2M, PRE |
| **GPCC** | PRE |
| **MERRA2** | T2M |
| **HIRS** | CLCT |
| **ISCCP** | CLCT |