# Peer review of "Exploring the Parameters Space of the Regional Climate Model COSMO-CLM 5.0 for the CORDEX Central Asia Domain"

_Geoscientific Model Development, 2020_

## Referee Comment (RC1) · Anonymous Referee #1 · 4 Aug 2020

**Review of https://doi.org/10.5194/gmd-2020-196**

**Exploring the Parameters Space of the Regional Climate Model COSMO-CLM 5.0 for the CORDEX Central Asia Domain**

by Emmanuele Russo et al.

**General comments**

After Russo et al. (2019) this is the second study with the overall goal to find a suitable setup for climate simulations, using the Regional Climate Model (RCM) COSMO-CLM (CCLM), for the CORDEX Central Asia Domain.

The present study creates and analyses a Perturbed Physics Ensemble to investigate the parameter space of CCLM for the Central Asia Domain in order to characterize the model parameter uncertainty and to determine the most sensitive parameters for the region, on which, and this seems to be the topic of a future study, the objective calibration method of Bellprat et al. (2012) will be apply.

Altogether, 26 parameters have been considered.

The study focuses on those model parameters that are essential for parameterized physical process, namely turbulence, land surface processes, convection, microphysics, radiation and process in the soil.

The study is carried out for the entire model domain but also for eleven sub-regions characterized by different climate conditions.

The model results are compared with observations for 2m temperature (T2M), precipitation (PRE), and total cloud cover (CLCT). The observational data are from three different sources for each variable. This allows taking into account the uncertainty of observations.

The analyses of parameter uncertainties have been conducted by a Performance Index (PI) metric. PI represents a normalized multivariate root-mean-square error (RMSE), weighted over different sources of uncertainties (the monthly standard deviation of the internal variability of the regional model, the monthly standard deviation of the interannual variations of observations, and the monthly standard deviation of the observational error derived from different reference datasets) and averaged over the model variables, the considered regions and the months of a selected year. Model sensitivity to the variations of parameter values is quantified by a positive definite Performance Score (PS), which can be calculated from PI. Improvements of worsening of the performance of the different experiments compared to a reference simulation is determined by a Skill Score (SS)

The results show that the variations of only a sub-set of the considered parameters are accompanied by relevant changes in model performances. But when considering

the different sub-regions these changes are not consistent; the model may show an opposite behaviour among different regions. A result, which could be expected considering the large size of the entire model domain and the different climate conditions prevailing in the sub-regions.

From this point of view the results of the transfer of the model setup to EURO-CORDEX region are also not surprising. They show that the sensitivity of the model to parameters perturbation for Central Asia is different than the one observed for Europe.

The present study is an important contribution demonstrating that an RCM has to be re-tuned, and its parameter uncertainty properly investigated, when setting up model experiments to different domains of study. As the authors emphasize, this is of importance in order to strengthen confidence in climate projections.

From this point of view, the study is scientifically significant.

**I recommend the publication of the study after some revisions (see specific comments below)**

**Specific Comments:**

**Page 4 line 9**: Panitz et al. (2014) describes an evaluation simulation forced by ERA-Interim, not a future projection study; cite Dosio et al. (2015) and/or Dosio and Panitz (2016) instead.

Dosio et al. (2015): Dynamical downscaling of CMIP5 global circulation models over CORDEX-Africa with COSMO-CLM: evaluation over the present climate and analysis of the added value. Clim Dyn 44, 2637–2661 (2015). https://doi.org/10.1007/s00382-014-2262-x

Dosio, A. and H.-J. Panitz (2016): Climate change projections for CORDEX-Africa with COSMO-CLM regional climate model and differences with the driving global climate models. Clim Dyn 46, 1599–1625 (2016). https://doi.org/10.1007/s00382-015-2664-4

Page 5, line 21: Zhang et al (2004) cited, but reference is missing

**Pages 5 and 6, section 2.3**: which spatial resolution did you use for the comparisons between model data and observations? I assume 0.5°. Please, mention it and say why you chose the specific spatial resolution and how you did the remapping.

**Page 8, line 9**: any idea why PS is lower for PRE than for T2M and CLCT? Just indicating this fact is not very satisfying.

**Page 8, line 10**: must be Tab. 2, not Tab. 3

**Page 8, section 3.1**: altogether, 9 parameters have been selected, which are recommended to conduct the objective calibration procedure following Bellprat et al

(2012). These 9 parameters are the 7 most sensitive parameters that show largest variation in PS, and in addition, two further, namely uc1 and soilhyd, which have been selected from the interpretation of PS dependency on each variable. Why not also rat_lam and tur_len being characterized, like uc1 and soilhyd, as "parameters with particularly small variations in PS calculated for single variables …" (see Page 8, line 5). To my opinion, especially the tur_len values ≥ 500 m are too high, and the smaller value shows slight improvements for CLCT and PRE. Baldauf et al. (2011) also demonstrated the sensitivity of results of NWP to the values of tur_len with improvements using smaller values, even smaller than the lower limit of 100 m used here.

I recommend considering at least also tur_len in a subsequent objective calibration study.

**Page 9, line 13**: must be "**c_lnd**", not "c_land"; delete the "a"

**Page 9, line 15**: for example here the authors assign the model bias, here with respect to T2M, to "structural problems in the model formulation". But what's about the quality/reliability of observations in such sub-regions like those representing Siberia? I would expect at least a short paragraph in the manuscript discussing this aspect. I cite:

"As models are frequently tuned on the basis of observational data, misguided model development can easily result from not taking into account observational uncertainties. For example, tuning models to observations in regions where the mean model bias strongly depends on the selected observational data set (e.g. in Norway) can deteriorate the model performance." These are the first two sentences of the Conclusions from a publication of Prein and Gobiet (2017) that perfectly describes the impacts of uncertainties in observations on regional climate analysis.

**Page 11, section 3.4**: I assume that the PS analysis has been performed for T2M, PRE, and CLCT together. This is not mentioned in the text.

**Page 11, line 8**: please explain why you only used the parameters e_surf, rlam_heat, rat_sea, and entr_sc for the transferability study. I would have expected that you would have considered also **qi0**, **uc1, fac_rootdp2.** With e_surf and qi0 you then would have considered the two parameters that you identified as those with "the largest effect on model performance", as you state in your Conclusions. Furthermore, rlam_heat, rat_sea, entr_sc, qi0, uc1, and fac_rootdp2 are those parameters that had been considered by Bellprat et al (2012) in their objective calibration study. This would, perhaps, give the opportunity for some comparative discussions on the results achieved for corresponding parameters.

**Comments Figures**:

**Figure 3**:  please, indicate in the caption that the red marker represent the PS values for the default values of the tested parameters (see also Table 2)

**Figures 5, 6, and 7**: it would be of advantage for the reader to group the experiments carried out in this study according to the physical processes the respective parameters are assigned to (as you did in in Table 2). It would be much easier for the reader to follow the discussions in the text also in the figures Example: on page 8, line 32, the authors describe, for T2M, changes in model performance over the Tibetian Plateau due to value variations of the surface parameters e_surf and pat_len. In Fig. 5 the reader finds the results for pat_len in the upper part, those for e_surf nearly at the end. This makes it hard to "synchronize" a discussion/interpretation in the text with the corresponding visualization in the figure.

**Comments Tables**:

**Table 2**: column "Description" for rlam_heat: missing "n" in the word "boundary"

**Additional Recommendation**: Russo et al (2019) investigated the sensitivity of CCLM results to different physical parameterizations. The simulations had been carried out also for the Central Asia Domain. The model version used (COSMO-CLM 5.0_clm9) offers the possibility to choose also different parameterization for bare soil evaporation. But this process had not been considered in Russo et al. (2019). But I could imagine that the process of bare soil evaporation could be important especially for a domain like Central Asia. Here I would like to refer to a study by Schulz and Vogel (2020) that demonstrated the positive impact of the resistance formulation of bare soil evaporation, which can also be chosen in COSMO-CLM 5.0_clm9. Therefore, I recommend that the authors repeat the simulation described in Russo et al (2019) with different parameterizations of bare soil evaporation, at least that one using the resistance formulation, analyse the results, and change their reference setup accordingly, if they find a positive impact, before they start with the actual objective calibration of model parameters.

**Literature**:

Baldauf et al. (2011): Operational Convective-Scale Numerical Weather Prediction with the COSMO Model: Description and Sensitivities. Monthly Weather Review, 139, 3387-3905, DOI: 10.1175/MWR-D-10-05013.1

Bellprat et al. (2012): Objective calibration of regional climate models. JOURNAL OF GEOPHYSICAL RESEARCH, VOL. 117, D23115, doi:10.1029/2012JD018262, 2012

Dosio et al. (2015): Dynamical downscaling of CMIP5 global circulation models over CORDEX-Africa with COSMO-CLM: evaluation over the present climate and analysis of the added value. Clim Dyn 44, 2637–2661 (2015). https://doi.org/10.1007/s00382-014-2262-x

Dosio, A. and H.-J. Panitz (2016): Climate change projections for CORDEX-Africa with COSMO-CLM regional climate model and differences with the driving global climate models. Clim Dyn 46, 1599–1625 (2016). https://doi.org/10.1007/s00382-015-2664-4

Prein, A. and A. Gobiet (2017): Impacts of uncertainties in European gridded precipitation observations on regional climate analysis. Int. J. Climatol.37: 305–327 (2017), DOI: 10.1002/joc.4706.

Russo et al. (2019: Sensitivity studies with the Regional Climate Model COSMO-CLM 5.0 over the CORDEX Central Asia Domain. Geosci. Model Dev. Discuss., https://doi.org/10.5194/gmd-2019-22.

Schulz, J.-P.and g. Vogel (2020): Improving the Processes in the Land Surface Scheme TERRA: Bare Soil Evaporation and Skin Temperature. Atmosphere 2020, 11, 513; doi:10.3390/atmos11050513

---

## Referee Comment (RC2) · Andreas Dobler (Referee) · 14 Aug 2020

Review of "Exploring the Parameters Space of the Regional Climate Model COSMO-CLM 5.0 for the CORDEX Central Asia Domain" by Emmanuele Russo et al.

General comments The study of Russo et al. is investigating the impact of different parameter settings in the COSMO-CLM RCM on the model's performance in the CORDEX region Central Asia Domain. It is using a PPE (Perturbed Physics Ensemble) setup, trying to answer the question on how (and whether) an optimal configuration can be found and if it is transferable to other regions (specifically, CORDEX Europe) and universal throughout the simulation domain. The most sensitive parameters are highlighted for three different variables (temperature, precipitation and total cloud cover) separately and commonly.

The study gives a very valuable insight into the parameter space for the different parameterisations available in COSMO-CLM and how they might influence the model results overall and for different variables and areas. The performance of the model is summarised in several metrics which are easy to understand.

The results of the study show that the sensitivity and performance of the model for the investigated parameter space is different in Central Asia and Europe. Also for sub-regions this can differ significantly and the results show opposite effects of several parameter settings for different areas. Thus, it shows that re-tuning the model and investigating the sensitivity when moving to a new model domain should be carried out, and also variations within the domain and for different variables should be considered.

Overall, the study represents a substantial contribution to modelling science and I recommend its publication. Some few specific comments for minor revision can be found below.

Specific Comments: Page 5 line 3: Normally, ERAInterim reanalysis data are used to drive RCMs evaluation and calibration experiments. Conversely ... –> "Normally" and "Conversely" are true for CORDEX simulations but I wouldn't use them as a general standard. I think you have a valid point there on the resolution jump. Thus, I'd suggest to write: "Within CORDEX, ERAInterim reanalysis data are used to drive the RCMs evaluation experiments and usually for calibration. NCEP2 data are employed in this study with the specific purpose of reproducing the spatial resolution jump ..."

Page 5 line 30: A k-means clustering technique (Steinhaus, 1956; Ball and Hall Dj, 1965; MacQueen et al., 1967; Lloyd, 1982; Jain, 2010; Russo et al., 2019) –> Do you really need to include all 6 references for the k-means clustering technique here?

Page 12 line 33: In this case, the reason for the biases is most likely related to some

structural error in the model formulation. –> I suggest adding "or the model setup", e.g. the horizontal and vertical resolution, rdheight, number of vertical levels or - for the IMO region - the proximity of the domain boundary could also be a reason for (parts of) the bias.

Please add a paragraph (either in 2.2 on observations or in the conclusions) on the uncertainty of the observation. Although you are using different data products, the source behind them is (at least for those based on station data) probably similar and may be sparse for some areas you consider.

Comments Figures and Tables: Additionally to the comments in RC1 (especially sorting the lines in figures 5-7), I'd suggest to increase the font size in figure 11 if possible.

---

## Author Comment (AC1) · 29 Sep 2020

Reply to
**Anonymous Referee**
*Russo, E., Soerland, S.L., Kirchner, I., Schaap, M., Raible, C.C. and Cubasch, U.:*
**Exploring the Parameters Space of the Regional Climate Model COSMO-CLM 5.0 for the CORDEX Central Asia Domain, Geosci. Model Dev. Discuss.,**
*https://doi.org/10.5194/gmd-2020-196.*

Dear reviewer,

Thank you very much for your effort in reviewing our paper.

Below we go point by point through your technical corrections, presented in *italic*, detailing how we dealt with your concerns reported in **bold**. Thank you.

*General Comments*

- *Page 4 line 9: Panitz et al. (2014) describes an evaluation simulation forced by ERA-Interim, not a future projection study; cite Dosio et al. (2015) and or Dosio and Panitz (2016) instead.*

  - Dosio, A. and H.-J. Panitz (2016): Climate change projections for CORDEX-Africa with COSMO-CLM regional climate model and differences with the driving global climate models. - *Dosio et al. (2015): Dynamical downscaling of CMIP5 global circulation models over CORDEX-Africa with COSMO-CLM: evaluation over the present climate and analysis of the added value. Clim Dyn 44, 26372661 (2015).*

  - *Dosio, A. and H.-J. Panitz (2016): Climate change projections for CORDEX-Africa with COSMO-CLM regional climate model and differences with the driving global climate models.*

  **We will correct the previous reference taking into account the new ones suggested by the referee.**

- *Page 5, line 21: Zhang et al (2004) cited, but reference is missing*

  **We will introduce the missing reference in the reference list.**

- *Pages 5 and 6, section 2.3: which spatial resolution did you use for the comparisons between model data and observations? I assume 0.5°. Please, mention it and say why you chose the specific spatial resolution and how you did the remapping.*

We conducted our analyses considering a spatial resolution of 0.5°for the comparison between model data and observations. Prior to the calculation of the considered metrics the model data were remapped onto the grid of the CRU dataset. For temperature we used a linear remapping, while for precipitation and cloud cover a conservative interpolation approach was employed. Following the comment of the reviewer we realized that such information is missing in the manuscript and we will provide it in section **2.3** of the new manuscript, where the analysis methods and metrics are discussed.

- *Page 8, line 9: any idea why PS is lower for PRE than for T2M and CLCT? Just indicating this fact is not very satisfying.*

  The value of PS is particularly low for precipitation because of higher biases with respect to the values of the uncertainties in this case. On the other hand, biases for T2M and CLCT are more in the range of the corresponding uncertainties. This is evident from Fig. 9 and is discussed in section 3.3, where we analyze the role of different uncertainties on the computation of the considered metrics. In the new version of the manuscript we will explain at the end of line 9 in page 8 that more analysis on this point will be introduced in the following sections. Also, we will try to extend the discussion in section 3.3. concerning the role of different uncertainties on the considered metrics.

- *Page 8, line 10: must be Tab. 2, not Tab. 3*

  Will be corrected accordingly.

- *Page 8, section 3.1: altogether, 9 parameters have been selected, which are recommended to conduct the objective calibration procedure following Bellprat et al(2012). These 9 parameters are the 7 most sensitive parameters that show largest variation in PS, and in addition, two further, namely uc1 and soilhyd, which have been selected from the interpretation of PS dependency on each variable. Why not also rat lam and tur len being characterized, like uc1 and soilhyd, as parameters with particularly small variations in PS calculated for single variables ... (see Page 8,line 5). To my opinion, especially the tur len values $\geq$ 500 m are too high, and the smaller value shows slight improvements for CLCT and PRE. Baldauf et al. (2011) also demonstrated the sen-*

*sitivity of results of NWP to the values of tur len with improvements using smaller values, even smaller than the lower limit of 100 m used here. I recommend considering at least also tur len in a subsequent objective calibration study.*

**Despite acknowledging the importance of additional parameters as suggested by the reviewer, our a priori decision was to select a maximum of 2 parameters for each of the model physical scheme. This choice was deliberately made for keeping the "costs" of a possible calibration procedure limited. Following the reviewer comment, we will make clearer in the new version of the manuscript the reasons for our decision. At the same time, we will try to highlight the fact that other parameters such as tur_len play an important role. Concerning the parameter rat sea, instead, we do not agree with the referee on the fact that it plays such an important role for the region.**

- *Page 9, line 13: must be $c_l nd, not c_l and; delete the a$*

  **Will be corrected.**

- *Page 9, line 15: for example here the authors assign the model bias, here with respect to T2M, to structural problems in the model formulation. But whats about the quality/reliability of observations in such sub-regions like those representing Siberia? I would expect at least a short paragraph in the manuscript discussing this aspect. I cite: As models are frequently tuned on the basis of observational data, misguided model development can easily result from not taking into account observational uncertainties. For example, tuning models to observations in regions where the mean model bias strongly depends on the selected observational data set (e.g. in Norway) can deteriorate the model performance. These are the first two sentences of the Conclusions from a publication of Prein and Gobiet (2017) that perfectly describes the impacts of uncertainties in observations on regional climate analysis.*

  **The considered metrics take already into account different sources of uncertainties in their definition, among which the one related to the use of different observational data-sets and the one related to the interannual variability of the reference observational data-set. For the case of T2M over Siberia (here SAR, CSA, DSS of Fig. 2 of the manuscript) Russo**

et al. 2019 showed that the model presents a remarkable warm bias in winter over Siberia. Despite the fact that over some point this large bias is associated with high uncertainty in observational data-sets, the comparison against different observations confirmed its sign and pattern: there is surely some problem for the model over this region. The issue would eventually be how to accurately assess the magnitude of this bias over the entire points of the region. The fact that all the perturbed parameters do not show significant improvements in simulated temperatures over the area (SS derived from PI), and a consequent reduction of the bias, is indicative of the fact that the model is likely missing or not accurately reproducing processes important for the region. It has to be acknowledge though that one possible reason for the high values of PI and its small variations when perturbing parameter values is that these changes are dumped by higher values of the uncertainties with respect to the bias over the region. Therefore for a better interpretation of our results we decided to include a section in the paper, discussing the role of uncertainties on the calculated metrics. From Fig. 9 of the former version of the manuscript it is possible to see that the role of the different uncertainties compared to the bias is relatively small for the 3 subdomains of Western Siberia, for almost all the months. This supports the idea that for the region, the model bias does not change significantly when changing parameter values. There is an underlying reason for the evinced biases that could possibly be reconducted to model formulation. As already said before, we acknowledge the fact that the discussion on the effect of the different uncertainty sources as presented in the former version of the manuscript could sensibly be improved and extended. We will try to do so in the new version of the manuscript. Additionally, considering the 2nd reviewer comment, we realized that in our previous conclusions we did not give enough weight to the fact that the evinced model sensitivity might change when changing the model setup, for example changing the size of the domain or the model resolution. We will try also to consider this point in the new version of the manuscript, when referring to errors in the model formulation.

- *Page 11, section 3.4: I assume that the PS analysis has been performed for T2M, PRE, and CLCT together. This is not mentioned in the text.*

  **Following the referee comment we realized that we did not specify how the PS analysis is conducted in section 3.4. We will specify it in the new version of the manuscript.**

- *Page 11, line 8: please explain why you only used the parameters e surf, rlam heat, rat sea, and entr sc for the transferability study. I would have expected that you would have considered also qi0, uc1, fac rootdp2. With e surf and qi0 you then would have considered the two parameters that you identified as those with the largest effect on model performance, as you state in your Conclusions. Furthermore, rlam heat, rat sea, entr sc, qi0, uc1, and fac rootdp2 are those parameters that had been considered by Bellprat et al (2012) in their objective calibration study. This would, perhaps, give the opportunity for some comparative discussions on the results achieved for corresponding parameters.*

  **We actually selected a priori 2 parameters for which the model seems to be particularly sensitive over the Central Asia domain and 2 for which it is not. In our opinion, seeing that some parameters that are not sensitive in one case are sensitive in the other, is already sufficient for supporting the hypothesis that calibration analyses should be performed when changing the domain of study. In this sense, according to the evinced results, we do not think it is necessary to perform further tests.**

*Comments Figures*

- *Figure 3: please, indicate in the caption that the red marker represent the PS values for the default values of the tested parameters (see also Table 2)*

  **will be corrected in the new version of the manuscript.**

- *Figures 5, 6, and 7:It would be of advantage for the reader to group the experiments carried out in this study according to the physical processes the respective parameters are assigned to (as you did in in Table 2). It would be much easier for the reader to follow the discussions in the text also in the figures Example: on page 8, line 32, the authors describe, for T2M, changes in model performance over the Tibetian*

*Plateau due to value variations of the surface parameters e surf and pat len. In Fig. 5 the reader finds the results for pat len in the upper part, those for e surf nearly at the end. This makes it hard to synchronize a discussion/interpretation in the text with the corresponding visualization in the figure.*

**We will try to sort the different experiments of Fig. 5-7 as suggested by the reviewer in the new version of the manuscript. On the other hand, concerning the referee comment on the discussion, we previously discussed the figures focusing on different regions and we would like to use the same approach in the new version of the manuscript. However, we will acknowledge the referee comment, reviewing the text for making the discussion more synchronized where necessary.**

---

## Author Comment (AC2) · 29 Sep 2020

Reply to
**2nd Referee**
*Russo, E., Soerland, S.L., Kirchner, I., Schaap, M., Raible, C.C. and Cubasch, U.:*
***Exploring the Parameters Space of the Regional Climate Model COSMO-CLM 5.0 for the CORDEX Central Asia Domain, Geosci. Model Dev. Discuss.,***
*https://doi.org/10.5194/gmd-2020-196.*

Dear reviewer,

Thank you very much for your effort in reviewing our paper.

Below we go point by point through your technical corrections, presented in *italic*, detailing how we dealt with your concerns reported in **bold**. Thank you.

*Specific Comments*

- *Page 5 line 3: Normally, ERAInterim reanalysis data are used todrive RCMs evaluation and calibration experiments. Conversely ... → Normally and Conversely are true for CORDEX simulations but I wouldnt use them as a general standard. I think you have a valid point there on the resolution jump. Thus, Id suggest to write: Within CORDEX, ERAInterim reanalysis data are used to drive the RCMs evaluation experiments and usually for calibration. NCEP2 data are employed in this study with the specific purpose of reproducing the spatial resolution jump .*

  **We agree and will modify the corresponding part of the text accordingly to the referee comment.**

- *Page 5 line 30:A k-means clustering technique (Steinhaus, 1956; Ball and Hall Dj,1965; MacQueen et al., 1967; Lloyd, 1982; Jain, 2010; Russo et al., 2019 → Do you really need to include all 6 references for the k-means clustering technique here?*

  **Here we could remove the reference of Russo et al. 2019 but, on the other hand, we would like to propose all the other references.**

- *Page 12 line 33: In this case, the reason for the biases is most likely related to some structural error in the model formulation. → I suggest*

*adding or the model setup, e.g.the horizontal and vertical resolution, rdheight, number of vertical levels or - for the IMO region - the proximity of the domain boundary could also be a reason for (parts of) the bias*

**We will follow the reviewer comment and try to highlight in the new version of the manuscript the fact that evinced model sensitivity might change when changing the model setup, for example for some areas close to the boundaries or the model resolution.**

- *Please add a paragraph (either in 2.2 on observations or in the conclusions) on the uncertainty of the observation. Although you are using different data products, the source behind them is (at least for those based on station data) probably similar and may be sparse for some areas you consider.*

**In the previous version of the manuscript we had a section (3.3) on the consideration of different uncertainty sources, where we discussed Fig. 8 and Fig. 9. More considerations on these figures could definitely be included in the new version of the manuscript. In particular, following the reviewer comment, we will try to add more details on the role of the observational uncertainties for the calculation of considered metrics over different regions.**

- *Comments Figures and Tables: Additionally to the comments in RC1 (especially sortingthe lines in figures 5-7), Id suggest to increase the font size in figure 11 if possible*

**We will try a new sorting for figures 5-7. At the same time, following the referee comment, we will increase the font of the axis in Fig. 11, more similarly to Fig.3 and 4.**